# Evaluation and Catheter Ablation of Ventricular Arrhythmias in Cardiac Sarcoidosis

**DOI:** 10.3390/jcm11226718

**Published:** 2022-11-13

**Authors:** Fatima M. Ezzeddine, Nicholas Tan, Konstantinos C. Siontis

**Affiliations:** Department of Cardiovascular Medicine, Mayo Clinic, Rochester, MN 55905, USA

**Keywords:** cardiac sarcoidosis, catheter ablation, ventricular arrhythmias

## Abstract

Ventricular arrhythmias are a common clinical manifestation in patients with cardiac sarcoidosis (CS) and other arrhythmogenic inflammatory cardiomyopathies (AIC). The management of sustained ventricular arrhythmias in these patients presents unique challenges. Current therapies include immunosuppressive, antiarrhythmic agents, and catheter ablation. Significant progress has been made in deciphering the importance of patient selection for ablation, systematic preablation evaluation, and optimal ablation timing, as well as ablation approaches and techniques. In this overview, we discuss the evaluation and management of ventricular arrhythmias in patients with CS, focusing on catheter ablation, which has evolved into an effective approach in reducing the burden of ventricular arrhythmias in these patients in the context of multifaceted treatment along with medical therapies.

## 1. Introduction

Arrhythmogenic inflammatory cardiomyopathies (AICs) are nonischemic cardiomyopathies that can result in ventricular arrhythmias due to myocardial inflammation [1]. AICs are most commonly caused by autoimmune disorders (such as sarcoidosis, eosinophilic myocarditis, giant cell myocarditis, chronic lymphocytic myocarditis, and other systemic autoimmune conditions with myocardial involvement), viral infections (such as enteroviruses, adenoviruses, parvovirus B19, human herpesvirus 6, and cytomegalovirus), and more rarely drug reactions [1]. 

Sarcoidosis is the most common of the AICs and represents a systemic inflammatory disorder due to an exaggerated immune response to unidentified antigens resulting in the formation of noncaseating granulomas and fibrosis [2]. Cardiac sarcoidosis (CS) can be associated with a poor prognosis and is present in up to one-third of patients with systemic sarcoidosis based on autopsy studies [3]. The main clinical manifestations of CS include conduction abnormalities, ventricular arrhythmias, heart failure, and sudden cardiac death [2]. In this review, we discuss the evaluation and management of ventricular arrhythmias in patients with CS. We specifically highlight the approach to, outcomes, and challenges of catheter ablation (CA) of ventricular arrhythmias in CS.

## 2. Pathogenesis of Ventricular Arrhythmias in Cardiac Sarcoidosis

CS can affect any of the heart layers: endocardium, myocardium, and/or epicardium. Cardiac involvement is usually multifocal with progression from active inflammation to fibrosis and scarring. The most commonly affected chamber is the left ventricle (LV) at the interventricular septum. CS has a predilection to the basal septum, lateral mitral annulus, and LV summit [4]. The right ventricle (RV) can also be involved in CS, and isolated RV sarcoidosis can mimic arrhythmogenic right ventricular cardiomyopathy (ARVC) [5]. Patients with CS are at an increased risk for ventricular arrhythmias, including premature ventricular complexes (PVCs), ventricular tachycardia (VT) (which can be monomorphic or polymorphic), and PVC-triggered ventricular fibrillation (VF) [6] (Figure 1). The arrhythmogenic substrate in CS can be located in either ventricle at any depth (subendocardial, mid-myocardial, and/or subepicardial). The His–Purkinje system can also be involved in ventricular arrhythmias in patients with CS [7]. 

The mechanisms of ventricular arrhythmias in CS include abnormal automaticity, triggered activity, and reentry (Figure 1) [8]. Areas of granulomatous inflammation can demonstrate abnormal automaticity. They can also disperse ventricular depolarization and repolarization resulting in reentrant arrhythmias [9]. Furthermore, healing of the sarcoid granulomas results in scar formation, which can mediate the formation of circuits for reentrant arrhythmias. Abnormal automaticity and triggered activity are more associated with the inflammatory phase, while reentry is more associated with the scar phase. The dominant substrate for VT in patients with CS is thought to be a postinflammatory scar from sarcoid granulomas [4,7]. It is worth noting that the underlying substrate for ventricular arrhythmias in CS is often heterogeneous and evolving, which makes treatment of ventricular arrhythmias more difficult with variable response to immunosuppression and antiarrhythmic medications.

## 3. Diagnostic Evaluation of Cardiac Sarcoidosis and Other Arrhythmogenic Inflammatory Cardiomyopathies

The diagnosis of CS can be challenging due to the overlap of symptoms with other cardiac AICs. Furthermore, CS is characterized by patchy disease distribution, which makes the diagnosis of CS difficult depending on the location and extent of cardiac involvement. A diagnosis of definite CS can be made by endomyocardial biopsy, which is particularly useful in patients with suspected isolated CS. However, the yield of endomyocardial biopsy is low, with a reported sensitivity in the 20–30% range [10]. The diagnostic yield of endomyocardial biopsy can be increased with the use of electroanatomic voltage mapping [11,12,13] and adjunctive use of substrate characterization with advanced cardiac imaging before biopsy [12].

Advanced cardiac imaging, including cardiac magnetic resonance (CMR) imaging and cardiac ^18^F-fluorodeoxyglucose positron emission tomography ^(18^F-FDG-PET), has been increasingly used in patients with suspected CS. On CMR imaging, late gadolinium enhancement (LGE) indicates inflammation or fibrosis/scar. The most common patterns of LGE in CS include subepicardial and mid-myocardial LGE along the basal septum and/or inferolateral wall [14]. In a meta-analysis including eight studies assessing the diagnostic accuracy of CMR imaging in patients with suspected CS, CMR imaging had a 95% sensitivity and 92% specificity for the detection of CS [15]. Recent data suggest that certain patterns of LGE may predict arrhythmic and heart failure outcomes in patients with CS [16].

Focal FDG uptake on cardiac PET imaging indicates disease activity in granulomatous myocarditis. In a meta-analysis including seven studies evaluating the accuracy of ^18^F-FDG-PET imaging for the diagnosis of CS, FDG-PET scan imaging had a sensitivity of 89% and a specificity of 78% for the detection of CS [17]. FDG-PET scan imaging is not only helpful in diagnosing CS but also in monitoring response to immunosuppressive therapy.

In recent years, the expanding use of metabolic activity FDG cardiac PET imaging and genetic testing in patients with ventricular arrhythmias has led to increased recognition of nonsarcoid, inherited cardiomyopathies presenting as AICs [18]. Traditionally, ARVC has been considered in the differential diagnosis of patients with ventricular arrhythmias and RV-dominant substrate. Differentiation from RV-dominant CS may be made based on electrocardiographic, imaging, invasive electroanatomic mapping criteria, and endomyocardial biopsy findings [19,20]. In patients with LV-dominant substrate, CS may be confused with left-dominant arrhythmogenic cardiomyopathy with a preponderance of desmoplakin (*DSP*) mutations, which can be found in around 50% of patients with ARVC, as well as acute and chronic myocarditis, and other familial dilated cardiomyopathies [21,22]. These conditions also present in relapsing and remitting phases, with increased myocardial cellular turnover in the acute phases represented as increased metabolic activity by FDG imaging, which can also be suggested by the clinical presentation. Differentiation of these AICs from CS is best achieved with histologic evaluation of affected myocardium.

## 4. Medical Management of Ventricular Arrhythmias

The primary treatment of CS with active inflammation is immunosuppression with corticosteroids. A typical regimen includes prednisone 30–40 mg daily for 4–8 weeks, followed by a taper by 5–10 mg every 2–4 weeks until reaching a maintenance dose of 5–15 mg daily. The maintenance dose is then continued for 9–12 months. Small series have also reported the effectiveness of intravenous methylprednisolone 40 mg once or twice daily for 3 to 5 days to control VT storm in patients with CS in addition to antiarrhythmic drug therapy [23]. Monitoring of treatment response with FDG-PET imaging is performed in intervals of 6–12 months after treatment initiation.

Steroid-sparing immunosuppressant agents such as methotrexate, leflunomide, mycophenolate mofetil, hydroxychloroquine, and azathioprine were previously considered second-line in the treatment of CS. However, most centers now initiate these agents early in the course of treatment in conjunction with corticosteroids in patients requiring long-term immunosuppression to reduce the side effects of corticosteroids [24]. The tumor necrosis factor-alpha antagonists such as infliximab and adalimumab are often reserved for refractory disease [25].

Padala and colleagues showed that early initiation of corticosteroids in CS was beneficial in preventing recurrence of ventricular arrhythmias as compared to delayed initiation of corticosteroids or failure to use corticosteroids [26]. Yodogawa and colleagues demonstrated that corticosteroids were more effective in reducing the burden of ventricular arrhythmias (PVCs and NSVT) in patients with less advanced left ventricular dysfunction (left ventricular ejection fraction (LVEF) ≥ 35%) as compared to patients with more advanced left ventricular dysfunction (LVEF < 35%) [27]. In patients with sustained monomorphic VT, Furushima and colleagues draw attention to the possibility of incessant VT emerging after initiation of steroid therapy due to abnormal automaticity [9]. Furthermore, the heterogeneous substrate in CS makes the response to steroid therapy variable. While steroids suppress inflammation in active granulomas, they may also result in more rapid scar formation and potential aneurysm formation, which can be the nidus for reentrant ventricular arrhythmias. Segawa and colleagues assessed the time course and factors associated with ventricular arrhythmias after introduction of steroid therapy in patients with CS and found that VT and electrical storm frequently occurred in the first 12 months after initiation of corticosteroid therapy [28].

There are limited data regarding the outcomes of antiarrhythmic medications in patients with CS. Class III antiarrhythmic agents, such as sotalol and amiodarone, are most commonly used in these patients. Amiodarone should be avoided in patients with advanced lung and/or liver disease due to sarcoid involvement, though it may be the only effective antiarrhythmic drug option in refractory arrhythmia cases. Class IC antiarrhythmic agents should generally be avoided in CS similarly to other structural heart diseases.

## 5. Implantable Cardioverter Defibrillator Therapy in Patients with Cardiac Sarcoidosis

Based on the 2017 American Heart Association (AHA)/American College of Cardiology (ACC)/Heart Rhythm Society (HRS) guidelines for management of patients with ventricular arrhythmias and prevention of sudden cardiac death, indications for implantable cardioverter defibrillator (ICD) implantation in patients with CS include: sustained VT or cardiac arrest (class I), LV dysfunction with an LVEF ≤ 35% (class I), or an LVEF > 35% with syncope, scar on CMR or PET scan, inducible ventricular arrhythmias, or indication for permanent pacing (class II a) [29]. Appropriate ICD therapies are common in patients, with CS with an estimated incidence rate of 15% per year [30]. Predictors of appropriate ICD therapies include left and right ventricular dysfunction, prior ventricular arrhythmias, and RV LGE [31,32,33].

## 6. Catheter Ablation for Ventricular Arrhythmias in Inflammatory Cardiomyopathies

Even with the use of immunosuppressive and antiarrhythmic medications, patients with inflammatory cardiomyopathy are still at increased risk for VT, which can lead to sudden cardiac death or ICD shocks. As such, CA is an important treatment modality for patients with PVCs and VT, especially when arrhythmias are refractory to medical therapy [8,34].

### 6.1. Inflammatory Substrate: Considerations for Ventricular Arrhythmia Ablation

Mechanisms for ventricular arrhythmias in CS can be broadly related to ongoing myocardial damage during the active phase, and fibrotic replacement of normal tissue and cardiac chamber remodeling facilitating reentrant circuits [6,34,35]. In a large study of patients with myocarditis, irregular and polymorphic ventricular arrhythmias were more prevalent among those with active disease [22]. This suggests a dynamicity to the arrhythmogenic substrate when inflammation is present. Furthermore, the risk of VT recurrence following CA was markedly higher among patients with active myocarditis [36]. This may relate to the evolving milieu that creates new substrate for VT initiation or maintenance. This trend is also seen in CS, which represents a distinct subset of myocarditis [37]. Thus, in the setting of active inflammation, it may be prudent to treat the underlying inflammation with immunosuppressive therapy and institute antiarrhythmic drug therapy first. However, in urgent scenarios of VT/VF storm or incessant VT in the presence of active inflammation, CA may still be effective in controlling the arrhythmia burden, and therefore, it should be considered as an adjunctive treatment modality in addition to immunosuppression [34].

Unlike ischemic cardiomyopathy, the arrhythmogenic substrate in AICs does not follow a coronary distribution [38]. The epicardium is often involved early in the disease course, leading some electrophysiologists to favor epicardial mapping and ablation upfront [36,39,40]. In a series of 127 patients with myocarditis undergoing VT ablation, 28 (22%) underwent combined endoepicardial ablation [36]. Mid-myocardial substrate is also important in the disease process, which adds to the complexity of the ablation procedures.

Efforts have been undertaken to elucidate the arrhythmogenic substrate in greater detail specifically in CS patients. Among 21 patients who underwent electroanatomic mapping at the time of VT ablation procedures, there was extensive scarring in both the RV (16/18) and LV (14/15) as well as the epicardium (7/8). The pattern of scarring in the RV appeared more confluent and evenly distributed across regions; by contrast, LV scarring was patchy and favored the septum, anterior wall, and outflow tract / aortomitral continuity [4]. In another study by Muser et al., 23 of 31 patients underwent preprocedural CMR imaging. The LV was more often involved than the RV (64% versus 36%, *p* = 0.013), with most patients exhibiting mid-myocardial (32%) or subepicardial (39%) LGE. Of note, 95% of patients were noted to have basal involvement. On electroanatomic mapping, bipolar (<1.5 mV endocardial, <1.0 mV epicardial) and unipolar (<8.3 mV in the LV, <5.5 mV in the RV) low-voltage abnormalities (LVAs) were recorded in both ventricles and the epicardium. A comparative analysis of imaging and electroanatomic mapping findings showed the best agreement between CMR imaging and endocardial unipolar mapping (κ = 0.4, *p* < 0.001) as well as CMR imaging and epicardial bipolar mapping (κ = 0.5, *p* < 0.001) [41]. In the multicenter study by Siontis et al., preprocedural CMR imaging was performed in 100 of 158 cardiac sarcoidosis patients undergoing VT ablation, of whom 88 (88%) patients exhibited LGE; among these, septal involvement (*n* = 64, 73%) was the most common although multifocal findings were also frequent (*n* = 55, 63%). Additionally, 31 (35%) and 19 (22%) had mid-myocardial and epicardial LGE, respectively [42]. It should be noted that RV LGE is associated with adverse outcomes in patients with CS [43]. RV involvement can present differential diagnostic challenges with ARVC. Basal septal involvement and more significant unipolar rather than bipolar voltage abnormality can support the diagnosis of sarcoidosis over ARVC [19]. Irrespective of the treatment strategy (medical therapy or CA), anteroseptal scar has been associated with higher recurrence rates of ventricular arrhythmias [44]. Overall, these studies provide insight into the complex mechanisms of ventricular arrhythmias in patients with CS.

### 6.2. Role of Preprocedure Imaging

Multimodality cardiovascular imaging techniques have become quintessential in the treatment of patients with suspected or confirmed AIC [45]. The utility of imaging prior to potential VT ablation lies in three main aspects: (1) Establishing the diagnosis, (2) Detecting active inflammation, and (3) Characterizing the arrhythmogenic substrate. Echocardiography is often first employed given its lower cost and widespread availability [45]. It is helpful in identifying myocardial or pericardial features concordant with AIC, including wall motion abnormalities, abnormal thinning/thickening, reduced ejection fraction, or dilated cardiac chambers. However, such findings may be absent in the early stages of disease and also lack specificity. Strain imaging may improve detection of AIC [46,47]. ^18^FDG PET/CT and CMR imaging have emerged as powerful and complementary tools in the assessment of AICs, especially CS [48,49,50]. In CS, disease stages can be inferred based on perfusion defects and/or pathologic FDG uptake (signifying inflammation) [49]. Furthermore, the presence of inflammation is an important predictor of VT recurrence following ablation [36,42]. Conversely, CMR imaging provides functional information (akin to echocardiography) as well as detailed tissue characterization [6,48,50]. Notably, identification of LGE indicates the presence of fibrosis or inflammation, which in turn harbor arrhythmogenic areas [37,41,42]. For these reasons, the systematic use of CMR with LGE imaging in patients with nonischemic cardiomyopathy, including likely inflammatory cardiomyopathies, may be associated with improved VT ablation outcomes [51].

### 6.3. Outcomes

Current data regarding VT ablation outcomes in nonsarcoid AICs are limited to observational studies. Although incompletely understood, the underlying etiology likely informs the approach and outcomes for VT ablation. In a series of 20 patients with biopsy-proven viral myocarditis and refractory VT, endocardial ablation was acutely successful in 14 (70%), whereas the other 6 (30%) underwent epicardial ablation 2–27 days (median 10 days) following the prior endocardial procedure; 18/20 (90%) were free of sustained VT over a median follow-up time of 28 months [39]. Combined endoepicardial mapping and ablation strategy was predominantly pursued in another cohort of patients with VT and a history of myocarditis (23/26, 88.5%); over a median follow-up period of 23 months, 20/26 (77%) remained VT recurrence free [40]. In the largest study to date, Peretto et al. reported VT ablation outcomes among 125 patients with biopsy- or imaging-confirmed myocarditis. Epicardial ablation was performed in most patients, with 90 (72%) undergoing epicardial-only ablation and 28 (22%) undergoing endoepicardial ablation; post-ablation, the clinical VT was noninducible in 119/125 (95%). Over a median follow-up period of 63 months, 82 patients (66%) were free of VT recurrence; this was dependent on the stage of myocarditis, whereby patients with active myocarditis were at higher risk of recurrence compared to those with previous (resolved) myocarditis [36]. Overall, these studies suggest that CA is effective in managing VT in myocarditis when: (1) An epicardial approach is undertaken; (2) Ablation is avoided during the active phase of myocarditis (when possible).

CA of ventricular arrhythmias in CS has been a topic of great interest in the past 1–2 decades [52]. The reported success rate of VT ablation in CS has been mixed, but it is generally lower than that in other forms of nonischemic substrates, including dilated cardiomyopathy or viral myocarditis [38]. Data from observational studies pertaining to CS-related VT ablation outcomes are summarized in Table 1 [4,7,37,42,53,54,55]. Most patients in these studies had reduced LVEF and suffered refractory arrhythmias despite antiarrhythmic and/or immunosuppressive medical therapies. Most patients had multiple VTs induced, both clinical and nonclinical. There was some variation in ablation targets across studies. For example, Jefic et al. identified a peritricuspid VT circuit in 4/9 patients [53], whereas the majority of VTs in Dechering et al. were ablated in the RV apex [54]. In two subsequent larger series, ablation was performed in the interventricular septum for 16/31 [41] and 9/24 [37] patients, respectively. Although the majority of VTs were due to automaticity or scar-related reentry, Naruse et al. also identified five patients with VT triggered by Purkinje fiber activity [7]. Additionally, a substantial subset of patients underwent epicardial mapping and ablation in the published CS ablation series [4,37,41,42,53,55]. Despite extensive ablation performed in these studies, noninducibility post-ablation ranged widely from 12.5 to 80.6%. Long-term follow-up periods again varied significantly across the studies. In a meta-analysis of 5 studies (totaling 83 patients), 45 patients (54.2%) had a VT recurrence [56]. In the largest multicenter registry investigation to date (*n* = 158), the 1- and 2-year cumulative incidences of VT post-ablation were 37% and 44%, respectively [42]. However, it is noteworthy that the VT burden was reduced significantly with CA. A compromised LV function (LVEF < 50%) and presence of inflammation by FDG-PET imaging before the CA were associated with lower survival free of VT recurrence, heart transplantation, or death (Figure 2).

### 6.4. Catheter Ablation in Ventricular Tachycardia Storm

Patients with inflammatory cardiomyopathy are at risk of malignant ventricular arrhythmias, notably VT/VF storm (defined as ≥3 sustained episodes within a 24 h period) and incessant VT (hemodynamically stable VT lasting >1 h) [6,36,57]. CA is often considered during such life-threatening presentations alongside aggressive medical treatment [34]. In one study, nine patients experienced VT storm, of whom seven were successfully treated with ablation [4]. Additionally, Siontis et al. found that, among 65 patients with a preablation history of VT/VF storm or incessant VT, 53 (82%) remained free of VT/VF storm or incessant VT following ablation [42]. Although preliminary, these data suggest that CA may play an important role in reducing the risk of recurrent malignant tachycardias in this patient population.

## 7. Conclusions

Ventricular arrhythmias in the setting of AICs, most often CS in clinical practice, represent unique management challenges requiring a multifaceted approach with both medical and ablative interventions. CA is an effective treatment modality in conjunction with antiarrhythmic drugs and immunosuppressive agents. Outcomes of CA are superior when performed earlier in the disease course before LV function has been compromised and when performed in the absence of active myocardial inflammation.

## 8. Summary Points

The diagnostic evaluation of patients with suspected cardiac sarcoidosis requires a thorough assessment for exclusion of mimicking conditions that may also present with inflammation by FDG-PET scan.Catheter ablation is an effective strategy for patients with inflammatory cardiomyopathy and ventricular arrhythmias (especially VT/VF storm) that are refractory to immunosuppression and antiarrhythmic drugs.Preprocedural cardiovascular imaging with ^18^FDG PET/CT and/or CMR imaging can be helpful in assessing the presence of inflammation and characterizing the involved substrate.When possible, catheter ablation should be deferred until active inflammation has been suppressed.

## Figures and Tables

**Figure 1 jcm-11-06718-f001:**
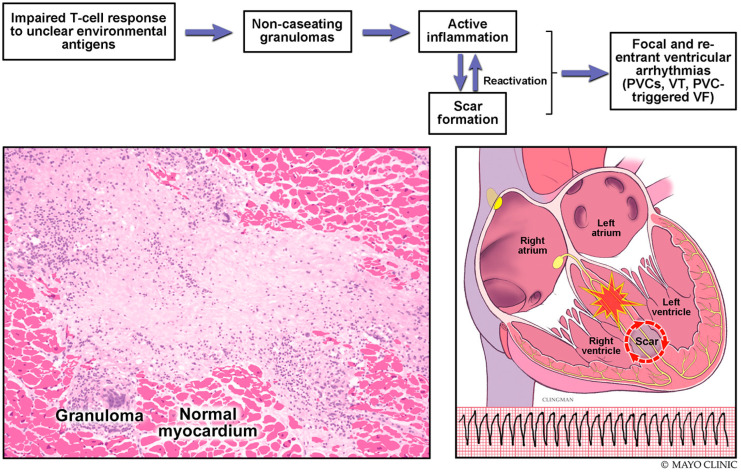
Pathogenesis of sustained ventricular arrhythmias in cardiac sarcoidosis.

**Figure 2 jcm-11-06718-f002:**
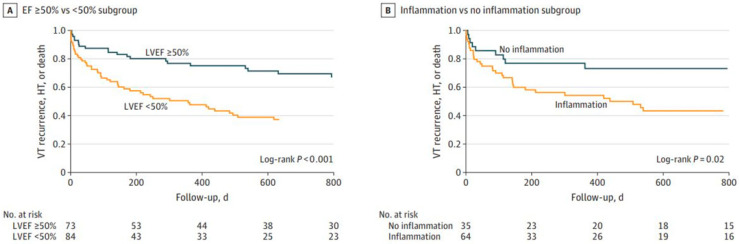
Outcomes of VT ablation in CS stratified by baseline (**A**) LVEF ≥50% vs. <50% and (**B**) presence vs. absence of active myocardial inflammation. Kaplan–Meier survival curves for the composite outcome of VT recurrence, heart transplant, or death. Reproduced with permission from Siontis et al. [42].

**Table 1 jcm-11-06718-t001:** Studies of catheter ablation for ventricular tachycardia in cardiac sarcoidosis.

Study	Cohort Size	LVEF	Mapping/Ablation Approach	Number of VTs Induced	Ablation Location(s)	Noninducibility Post-Ablation (%)	Follow-Up	VT Recurrence on Follow-Up
Koplan 2006 [55]	8	34 ± 15%	Substrate, pace map, entrainment	4.0 ± 2.0	Endocardial LV/RV; epicardial (*n* = 2)	1 (12.5%)	6 months to 7 years	4 (50%)
Jefic 2009 [53]	9	42 ± 14%	Substrate, pace map, entrainment, activation	4.9 ± 6.6	LV (*n* = 3), RV (*n* = 5), both (*n* = 1); epicardial (*n* = 1); peritricuspid area (*n* = 4)	5 (62.5%)	4 to 52 months	3 (37.5%)
Dechering 2013 [54]	8	36 ± 19%	Substrate, pace map, entrainment, activation	3.7 ± 1.9	Endocardial LV/RV, majority at RV apex	5 (62.5%)	6 months	1 (12.5%)
Naruse 2014 [7]	14	40 ± 12%	Substrate, pace map, entrainment, activation	4.1 (NR)	Endocardial LV/RV, targeting scar- (*n* = 13) or Purkinje-related (*n* = 5) VT; no epicardial map/ablation	NR	Median 33 months (IQR 26–46)	6 (42.9%)
Kumar 2015 [4]	21	36 ± 14%	Substrate, pace map, entrainment, activation	4.7 (NR) *	Endocardial LV/RV; epicardial (*n* = 8)	9 (42.9%) after 1st procedure	Median 2 years (IQR 1–9.5)	18 (85.7%) after 1st procedure; 63% 1-year cumulative incidence post last procedure
Muser 2016 [41]	31	42 ± 15%	Substrate, pace map, entrainment, activation	Median 3 (IQR 1–5) *	Endocardial LV/RV; epicardial (*n* = 8); interventricular septum (*n* = 16)	25 (80.6%) after last procedure	Median 2.5 years (IQR 1.3–5.2)	29% and 45% 1- and 2-year cumulative incidence post last procedure
Kaur 2020 [37]	24	48 ± 13%	Substrate, pace map, entrainment, activation	2.7 ± 2.0	Endocardial LV/RV; epicardial (*n* = 10); interventricular septum (*n* = 9)	16 (66.7%)	Mean 5.7 ± 3.9 years	46% at end of follow-up
Siontis 2021 [42]	158	Median 45% (IQR 34–55)	Substrate, pace map, entrainment, activation	Median 3 (IQR 1–4)	Endocardial LV/RV; epicardial (*n* = 39)	85 (53.8%)	Median 2.5 years (IQR 1.1–4.9)	37% and 44% 1- and 2-year cumulative incidence post procedure

* Over multiple procedures. AAD—antiarrhythmic drug; NR—not reported.

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
