# Peer review of "Evaluation and Catheter Ablation of Ventricular Arrhythmias in Cardiac Sarcoidosis"

_jcm, 2022, doi:10.3390/jcm11226718_

Round 1

Reviewer 1 Report

Dr Ezzeddine and collegues shoul be congratuleted for the paper: "Evaluation and Catheter Ablation of Ventricular Arrhythmias in Cardiac Sarcoidosis and Other Arrhythmogenic Inflammatory Cardiomyopathies". This review is well written, addresses in detail and comprehensively the difficult topic of treating complex ventricular arrhythmias in patients with cardiac sarcoidosis.

Perhaps I would add a short paragraph on the use of ICD in these patients

I add only a few minor notes:

- in the abstract, line 31, I would put first "patient selection for ablation" and then "pre-ablation evaluation"; as the pre-ablation evaluation is done only after selecting patients who are candidates for such treatment

- In the abstract, line 34, I would remove "and other AICs" as the study is about ablation in pts with cardiac sarcoidosis

- the same in the Introduction, line 45 and 47

-  the same pag 8, line 156

- On page 5, line 102, I believe that the differential diagnosis between CS and ARVC also consists of imaging tests and biopsy findings

- On page 5, line 104, the DSP abbreviation of the desmoplakin gene should be written in italics (DSP) as usual for genes

- On page 9, line 175, I would also cite IV septal portions as a technically more difficult ablation site with less acute and long-term procedural success  (see "Prior myocarditis and ventricular arrhythmia: the importance of scar pattern. Heart Rhythm. 2021 Apr;18(4):589-596. doi: 10.1016/j.hrthm.2020.12.016.).

Author Response

Dr Ezzeddine and colleagues should be congratulated for the paper: "Evaluation and Catheter Ablation of Ventricular Arrhythmias in Cardiac Sarcoidosis and Other Arrhythmogenic Inflammatory Cardiomyopathies". This review is well written, addresses in detail and comprehensively the difficult topic of treating complex ventricular arrhythmias in patients with cardiac sarcoidosis.

Perhaps I would add a short paragraph on the use of ICD in these patients

Reply: Thank you for kind words. A paragraph about the use of ICD in patients with cardiac sarcoidosis was added (page 8)

I add only a few minor notes:

- in the abstract, line 31, I would put first "patient selection for ablation" and then "pre-ablation evaluation"; as the pre-ablation evaluation is done only after selecting patients who are candidates for such treatment

Reply: The suggested edits were made.

- In the abstract, line 34, I would remove "and other AICs" as the study is about ablation in pts with cardiac sarcoidosis

Reply: “and other AICs” was deleted from the abstract.

- the same in the Introduction, line 45 and 47

Reply: The suggested edit was made.

-  the same page 8, line 156

Reply: Done.

- On page 5, line 102, I believe that the differential diagnosis between CS and ARVC also consists of imaging tests and biopsy findings

Reply: Differentiation by imaging and endomyocardial biopsy findings was added.

- On page 5, line 104, the DSP abbreviation of the desmoplakin gene should be written in italics (DSP) as usual for genes

Reply: DSP was written in italics.

- On page 9, line 175, I would also cite IV septal portions as a technically more difficult ablation site with less acute and long-term procedural success  (see "Prior myocarditis and ventricular arrhythmia: the importance of scar pattern. Heart Rhythm. 2021 Apr;18(4):589-596. doi: 10.1016/j.hrthm.2020.12.016.).

Reply: The citation was added (reference 46).

Reviewer 2 Report

1. Please add in p.5, line 104 that arrythmogenic cardiomyopathy can be present in the left ventricle in up to 50% of the patients .

2. Please add in p.6 line 108 ... but can be also suggested by the clinical presentation .

3. Please do not use contraction , p 10 line 215.

4. Please define other arrythmogrenic inflammatory  cardiomyopaties ( myocarditis ? Churg Strauss ? ARVC ? Polyarteritis nodosa ? ) .

5.Please a summary table comparing occurence and efficacy of medical treatment/ablation  of cardiac sarcoidosis vs other inflammatory cardiomyopathy ( and define those cardiomyopathies ) .

Author Response

  1. Please add in p.5, line 104 that arrhythmogenic cardiomyopathy can be present in the left ventricle in up to 50% of the patients.

Reply: The suggested edit was made.

  1. Please add in p.6 line 108 ... but can be also suggested by the clinical presentation .

Reply: The suggested edit was made.

  1. Please do not use contraction , p 10 line 215.

Reply: This has been changed to “complexes”.

  1. Please define other arrhythmogenic inflammatory cardiomyopathies ( myocarditis ? Churg Strauss ? ARVC ? Polyarteritis nodosa ? ) .

Reply: Definition and causes of AICs were added to the introduction section.

5.Please a summary table comparing occurrence and efficacy of medical treatment/ablation of cardiac sarcoidosis vs other inflammatory cardiomyopathy ( and define those cardiomyopathies)

Reply: Data regarding medical treatment and ablation outcomes in patients with VT and most non-sarcoid AICs is limited to small patient series, therefore we feel it is difficult to draw meaningful conclusions regarding their comparative efficacy compared to CS. One exception is viral myocarditis where some evidence for catheter ablation has accumulated and this is discussed in the Outcomes section on page 12. Also, in response to the first reviewer’s comment, we have now de-emphasized the other AICs from our discussion and focus more on cardiac sarcoidosis in the revised manuscript. However, if the reviewer and editors feel that this is critical to the review content, then we could go back and build a table despite the limited data available in the literature.